# Genome-Wide Analysis of the *SPL* Gene Family and Expression Analysis during Flowering Induction in *Prunus* × *yedoensis* ‘Somei-yoshino’

**DOI:** 10.3390/ijms231710052

**Published:** 2022-09-02

**Authors:** Lan Gao, Tong Lyu, Yingmin Lyu

**Affiliations:** 1Beijing Key Laboratory of Ornamental Plants Germplasm Innovation & Molecular Breeding, China National Engineering Research Center for Floriculture, College of Landscape Architecture, Beijing Forestry University, Beijing 100083, China; 2Beijing Flower Engineering Technology Research Center, Plant Institute, China National Botanical Garden North Park, Beijing 100093, China

**Keywords:** *Prunus × yedoensis* ‘Somei-yoshino’, *SPL* transcription factor, flower induction, genome, expression analysis

## Abstract

SQUAMOSA Promoter-Binding Protein-Like (*SPL*) genes encode plant-specific transcription factors which bind to the SQUAMOSA promoter of the MADS-box genes to regulate its expression. It plays important regulatory roles in floral induction and development, fertility, light signals and hormonal transduction, and stress response in plants. In this study, 32 *PySPL* genes with complete *SBP* (squamosa promoter binding protein) conserved domain were identified from the genome of *Prunus × yedoensis* ‘Somei-yoshino’ and analyzed by bioinformatics. 32 *PySPLs* were distributed on 13 chromosomes, encoding 32 *PySPL* proteins with different physical and chemical properties. The phylogenetic tree constructed with *Arabidopsis thaliana* and *Oryza sativa* can be divided into 10 subtribes, indicating *PySPLs* of different clusters have different biological functions. The conserved motif prediction showed that the number and distribution of motifs on each *PySPL* is varied. The gene structure analysis revealed that *PySPLs* harbored exons ranging from 2 to 10. The predictive analysis of acting elements showed that the promoter of *PySPLs* contain a large number of light-responsive elements, as well as response elements related to hormone response, growth and development and stress response. The analysis of the *PySPLs* expressions in flower induction and flower organs based on qRT-PCR showed that *PySPL06**/22* may be the key genes of flower development, *PySPL01/06* and *PySPL22* may play a role in the development of sepal and pistil, respectively. The results provide a foundation for the study of *SPL* transcription factors of *Prunus × yedoensis* ‘Somei-yoshino’ and provide more reference information of the function of *SPL* gene in flowering.

## 1. Introduction

Transcription factor is a kind of DNA binding protein that binds to a specific sequence upstream of the 5′ end coding region to regulate its expression by altering the transcriptional level of the gene. According to the different region of the DNA-binding structure, transcription factors can be distinguished into families and subfamilies [1,2]. SQUAMOSA promoter-binding protein-like (*SPL*) transcription factor, or the SBP protein, is a kind of transcription factor ubiquitous in green plants with a highly conserved SBP domain. This domain contains two zinc-finger structures, C3H (C-C-C-H) and C2HC (C-C-H-C) [3], while the C-terminus contains a highly conserved bidirectional nuclear localization signal (NLS) [4]. This structural domain binds to the SQUAMOSA promoter of the MADS-box genes to regulate its expression [5]. *SPL* is a multigene family involved in plant growth, flower development, fertility, light signals and hormonal transduction and stress responses [6,7,8,9,10,11]. Since the discovery of the *SPL* genes from snapdragons, the *SPL* genes in *Gossypium* spp. [12], *Populus trichocarpa* [13], *Betula luminifera* [14], *Ziziphus jujuba* [15] and other plants have been identified successively.

*SPL* gene regulates the flowering time of plants. *AtSPL3*, *AtSPL4* and *AtSPL5* in *A**. thaliana* are involved in the photoperiodic pathway through *PNY* and *PNF* to promote flowering [16]. Over expression of *AtSPL3* and *VpSBP11* with high homology to *AtSPL3* in *Vitis vinifera* can activate the expression of *FRUITFULL(FUL)*, *APETALA1(AP1)* and *LEAFY(LFY)* genes to advance flowering time in *A**. thaliana* [17,18]. *SUPPRESSOR OF OVEREXPRESSION OF CONSTANS1(SOC1)* and *miR156-SPL3* regulatory modules interact in response to plant photoperiod and gibberellin signaling and thus participate in the control of flowering time in *A**. thaliana* [19,20,21]. *AtSPL9*, *AtSPL10* and *AtSPL15* are involved in the regulation of flowering time via the age pathway in *A**. thaliana*, while miR156/172 are the two miRNAs important in the age pathway. The miR156 delays flowering in *A**. thaliana* by suppressing *AtSPL15* activity [22], while *AtSPL9* and *AtSPL10* activate expression of miR172 to accelerate flowering [23]. miR156 in *Prunus*
*mume* regulates the flowering process of *P**. mume* by down- or up-regulating the expression of *SPL* genes [24].

*SPL* can also regulate floral transition, floral organ development and fertility. For example, *PhSPLs* in *P**. hybrida* was found to be mainly expressed in axillary buds and inflorescence, where *PhSPL9a* and *PhSPL9b* facilitated the transition from vegetative growth to reproductive growth [25]. Over expression of *AtSPL3* in *A**. thaliana* under long daylight can cause abnormal inflorescence development [17], and *OsSPL13*, which is homologous to *AtSPL3*, also play a role in regulating flower development in *Oryza sativa* [26]. Over expression of *AtSPL10* suppresses pedicel elongation in *A**. thaliana* [27]. The miR156-mediated *AtSPL2* in affects the development of petal and sepal, pollen production and viability in *A**. thaliana* [28].

Flowering cherry (*Prunus × yedoensis* ‘Somei-yoshino’) is a famous early spring ornamental flowering tree with beautiful plant shape, brilliant flower color and early flowering period; it has a high ornamental value and is widely planted in the world. It has been shown that the expression of *FLOWERING LOCUS T(FT)* and gibberellin-mediated *DELLA* are increased during floral development before flowering in *Prunus × yedoensis* ‘Somei-yoshino’ [29], while the role of *SPL* in floral formation determination and floral development in this plant has rarely been reported. Based on the genomic data of *Prunus*
*× yedoensis* ‘Somei-yoshino’ [29], 32 *SPL* genes were searched and identified. The chromosomal location, phylogenetic relationship, conserved domain, conserved motifs, gene structure and collinearity were carried out by bioinformatics methods. Finally, qRT-PCR was used to compare the expression levels of some *PySPLs* in different floral developmental stages and different floral organs, which provides some reference to further study the biological functions of *SPL* genes in floral transition and flower development in flowering cherry.

## 2. Results

### 2.1. Screening, Identification and Sequence Analysis of PySPLs

A total of 43 *SPL* genes were initially identified using BlastP with 16 *Arabidopsis thaliana* SPL protein sequences as the reference and screened out by Pfam (PF031100). Based on conserved structural domains followed by removal of redundancy, 32 *SPL* genes were finally obtained within the *Prunus × yedoensis* ‘Somei-yoshino’ genome, and respectively named *PySPL 01*~*PySPL 32* according to their location on the chromosome (Table 1). Physicochemical properties analysis showed that the number of amino acids encoded by *PySPLs* was between 152 and 1070, molecular weight between 17,505.46~118,346.45 kD, theoretical pl between 5.75 and 10.12, and aliphatic index between 36.17 and 78.9. All members were unstable and hydrophilic in nature except *PySPL11*. All members were predicted to be localized in the nucleus, except *PySPL04*, *PySPL19*, *PySPL20*, *PySPL22*, *PySPL23*, and *PySPL31* in the cytoplasm and nucleus.

### 2.2. Chromosomal Localization of PySPLs

The chromosomal mapping results (Figure 1) showed that 32 *PySPLs* were distributed on all the 13 chromosomes except chromosomes SPA3/SPA8/SPE4/SPE6/SPE8. Six *PySPLs* were distributed on the chromosome SPA0, SPA7/SPE3/SPE7 distributed the least members (1).

### 2.3. Phylogeny Analysis of PySPLs

A Neighbor-Joining (NJ) phylogenetic tree of the SPL protein was constructed by using the aligned protein sequence of all *PySPLs*, *AtSPL**s* and *OsSPL**s* (Figure 2, Appendix A). A total of 68 SPL proteins can be roughly divided into 10 clusters and 32 PySPLs are distributed in all 10 clusters. The cluster V contains the most members of PySPLs (5), cluster I, II, III, VI, VII, VIII contains the least members of PySPLs (2). Meanwhile, PySPLs proteins have higher similarity with AtSPLs proteins compared to OsSPLs proteins, which may be related to the fact that both *Prunus* × *yedoensis* ‘Somei-yoshino’ and *A**. thaliana* are dicotyledons, while *Oryza sativa* is monocotyledonous.

### 2.4. Multiple Sequence Alignment and Conserved Domain Visualization of PySPLs

Multiple sequence alignment and conserved domain visualization of the *PySPLs* showed that all PySPLs protein contain highly conserved *SBP* domains (Figure 3), including two zinc-finger structures (Cys-Cys-Cys-His, Cys-Cys-His-Cys) and one bidirectional nuclear localization signal (NLS).

### 2.5. Conserved Motifs and Gene Structure Analysis of PySPLs

A phylogenetic tree of 32 *PySPLs* was constructed to further analyze their evolutionary relationships (Figure 4A). Conserved motifs of all 32 *PySPLs* were analyzed by the online software MEME. The results showed that Motif 1 and Motif 2 are included in all *PySPLs* member, are functional conserved motifs of *PySPLs* and contain the SBP domain. Meanwhile, *PySPLs* with closer evolutionary relationships contain approximately the same conserved motif. Almost all *PySPLs* contained more than two conserved motifs except *PySPL01*, *PySPL12,* and *PySPL17* (Figure 4B). The number of exons and introns contained by *PySPLs* varied widely (Figure 4C). *PySPL20* and *PySPL22* had the longest gene sequences and contained the largest number of exons (10), while *PySPL01*, *PySPL06*, *PySPL11*, *PySPL12* and *PySPL17* contained the fewest number of exons (2). A total of 20 *PySPLs* have 3′-UTR and 5′-UTR; 11 members lack 3′-UTR and 5′-UTR; one member (*PySPL19*) lacks 5′-UTR. *PySPLs* with closer evolutionary relationships have similar gene structures.

### 2.6. Cis-Acting Elements Analysis of PySPLs Promoter

2000 bp upstream sequences of the start codon of 32 *PySPLs* contains a large number of light response elements, phytohormone (ABA, GA, Auxin, SA, Meja) response elements and some specific response elements including meristem expression, low-temperature induction, drought induction, anaerobic induction, circadian control, defense and stress response (Figure 5). Light response elements are distributed among all *PySPLs* and accounted for the largest number of total elements in each *PySPL*, which indicating that the expression of *PySPLs* is likely to be influenced by light; second, all *PySPLs* promoter regions contain anaerobic induction response elements except *PySPL12*.Phytochrome response element and seed-specific response element are only distributed in *PySPL14/30* and *PySPL06/11* promoter region, respectively. The above results indicated that *PySPLs* is regulated differently, suggesting the functions of *PySPLs* were diverse.

### 2.7. Collinearity Analysis of PySPLs

According to the collinearity analysis results (Figure 6), the *PySPL* gene family is formed mainly due to the segmental duplication of chromosomes. A total of 31 genes in the *PySPL**s* family except *PySPL01* were co-linearly related (26 pairs), and all of them were due to segmental duplications. There are cases where one *PySPL* gene is co-linearly related to multiple *PySPLs*, such as *PySPL08* being co-linearly related to *PySPL15*, *PySPL25* and *PySPL31*. *PySPL14* is co-linearly related to *PySPL23*, *PySPL28* and *PySPL30. PySPL01* did not have a collinearity relationship. Presumably it appeared before the differentiation of cherry species.

### 2.8. Expression Analysis of PySPLs in Flowering Induction

The morphology of the samples in each period (Figure 7) showed that, compared with the dormant period (DP), the flower buds of *Prunus* × *yedoensis* ‘Somei-Yoshino’ continued to expand, and the flower primordium continued to grow within one month before flowering. At 20 days before flowering (24DBF), it was clearly observed that each flower bract was within the bud and the anthers and ovary were within the flower bract. Ten days before flowering (14DBF), the sepals, petals, stamens and pistils were clearly visible in form and color, thus indicating that the development of floral organs was basically completed ten days before flowering. Meanwhile, the pedicel and calyx tube elongate significantly ten days before flowering. A total of 12 *PySPLs* containing representative conserved motif distributions and gene structures were selected for qRT-PCR based on their evolutionary developmental relationships (Figure 8). In contrast to dormant periods (DP), the relative expression level of some *PySPLs* was reduced significantly, including *PySPL02*, *PySPL0**8*, *PySPL**10* and *PySPL31*, which may have a negative regulatory effect on flower induction. The relative expression level of *PySPL01*
*and PySPL14* was not significantly different in the early stage and decreased at the basic completion of floral organ development (14DBF). *PySPL06* had the highest relative expression at mid-flower development (24DBF) and then decreased, considering that *PySPL**06* is associated with flower organ development in *Prunus* × *yedoensis* ‘Somei-yoshino’. *PySPL26* had the highest relative expression level in the progress of flower primordium (34DBF and 24DBF) while the relative expression level of *PySPL07*, *PySPL22**, PySPL29* and *PySPL32* changed at random in all periods (3DBF).

In addition, the relative expression of 12 *PySPLs* genes in different flower organs which were collected at the blooming stage (0DBF) was analyzed with the dormant flower buds (DP) as a control and the *ACTIN* gene as the internal reference by qRT-PCR (Figure 9). *PySPL01*, *PySP06*, *PySPL07* and *PySP31* had the highest relative expression in sepals, *PySPL22* and *PySPL29* had the highest relative expression in pistils and *PySPL14* was expressed in all organs with insignificant differences. The expressions of *PySPL08*, *PySPL10*, *PySPL26* were barely detected while it is detectable during flowering development. The expression of *PySPL32* was not detected while its expression in 3DBF is almost the same as DP (Figure 7).

## 3. Discussion

With the continuous development of genomics, more and more attention has been paid to whole-genome identification of transcription factors and their gene expression profiles. *SPL* is a plant-specific transcription factor, and the number of its members is independent of genome size [30]. Genome-wide identification of the *SPL* gene family has been reported in many species and is responsible for diverse physiological activities such as floral development, stress response and phytohormone transduction. However, no systematic study has been performed in flowering cherry. This study comprehensively identified the *SPL* genes of *Prunus* × *yedoensis* ‘Somei-yoshino’ and carried out a systematically analysis on them.

*PySPLs* were distributed on different chromosomes in *Prunus × yedoensis* ‘Somei-yoshino’ with widely different physicochemical properties, and the distribution was not significantly associated with gene similarity or covariance. The phylogenetic tree constructed by *PySPL**s*, *AtSPLs* and *OsSPL* could be divided into 10 subclades, among which eight subclades contained members of *AtSPLs,* which is the same distribution as the phylogenetic relationship of *AtSPLs* [31,32]. It indicates that most of the *PySPLs* members had high homology with *AtSPLs*. The conserved motifs and gene structures of members in the same subclade are roughly the same, and most of them were co-linearly related, *PySPL09*, *PySPL21* and *PySPL26* were distributed at different positions on different chromosomes with same conserved motif (Motif1/2/5/9/10) and similar gene structures. Meanwhile, different subclade members contained different conserved motifs, which may determine the diversity of functions of *PySPLs*. *PySPL06* and *PySPL11* clustered with *AtSPL3*, *AtSPL4* and *AtSPL5* in the same subclade, indicating their higher homology relationships, suggesting that *PySPL06*, *PySPL11* may be related to flowering regulation of *Prunus* × *yedoensis* ‘Somei-yoshino’.

Gene duplication is the basis of the functional differentiation of homologous genes, is the main pathway for the generation of new functional genes and plays a pivotal role in the evolution of species [33]. Gene duplication mainly includes segmental duplication, whole-gene duplication, tandem duplication and replicative transposition [34]. Collinearity analysis indicated that there was a large number of segmental replications in the *PySPLs*, suggesting that segmental replication is the main way in which the *PySPLs* family evolved and expanded. The differences in the number of coding regions of *PySPLs* may be related to the loss and mutation of segmental replication during evolution [35,36].

Promoter is a DNA region located upstream of the transcription start site that can be recognized and bound by RNA polymerase. It has a transcription start site and is enriched with conserved sequences of TA (TATA region) as well as CCAAT sequences (CAAT region) and other key elements that can improve the initiation efficiency. Promoter also has some specific elements such as light-responsive *cis*-elements and G-boxes that can bind to specific *trans*-acting elements to initiate target genes and express them at specific intensities in specific times and spaces [37]. 29% of the specific elements contained in the promoter region of *PySPLs*, were associated with hormone response, which also indicates a strong association of *SPL* genes with plant phytohormone response; 45% were associated with light response and the *FT/SOC1*-mediated photoperiodic pathway can promote flowering in *Arabidopsis thaliana* by binding to the *AtSPL3, AtSPL4* and *AtSPL5* promoter region, which dependent on *PENNYWISE* (*PNY*) and *POUND-FOOLISH* (*PNF*) to induce its expression [16,38,39,40]. The expression of *FT* in *Prunus*
*×*
*yedoensis* ‘Somei-yoshino’ flower buds increased and then decreased between the end of dormancy and flowering [29], and the expression trend was basically the same as that of *PySPL06*, which has high homology with *AtSPL3*, *AtSPL4* and *AtSPL5*, suggesting that the large number of light response elements contained in the promoter region of *PySPLs* may be related to *FT*-mediated photoperiod-induced flowering in *Prunus × yedoensis* ‘Somei-yoshino’ (Appendix A) [16,17,18,19,20,21].

*SPL* transcription factors have an important role in plant flowering, and they regulate the floral transition various Flowering induction pathway in plants [41,42,43]. The development of floral organs was largely completed by 10 days before flowering. During the previous time, the expression of most *PySPLs* gradually decreased, among which *PySPL0**6* and *PySPL22* had a significant increase in the middle and then decreased. NCBI-BlastP compared the three genes with *Arabidopsis thaliana* and found that *PySPL06* had the highest homology with *AtSPL**3*, which was the same as the conclusion of previous studies [17], indicating that *PySPL06* may be involved in the regulation of flower organs development in flowering cherry; *PySPL22* had the highest homology with *AtSPL7* and was very prominently expressed in the pistil, which is presumed to be involved in pistil development. Furthermore, miR156 can delay plant flowering by suppressing *SPL* expression [22], however, the expression of some *PySPLs* gradually decreased during the flowering induction process, such as *PySPL**08*, *PySPL10* and *PySPL31*, which is in contrast to previous studies and needs further investigation.

The extent of flower organ growth depends on cell proliferation and cell expansion, and these two processes coordinate with each other to determine the size of the entire floral organ of the plant [44,45]. Over expression of miR156 increases cell size and decreases cell number, whereas over expression of *AtSPL3*, *AtSPL4*, *AtSPL5* and *AtSPL15* decreases cell size and increases cell number [46]. To further understand the role of these *PySPLs* in floral organ development, the expression of these *PySPLs* in different floral organs was analyzed. The final development of sepal may be related to *PySPL01* and *PySPL06*, *PySPL22* may be strongly associated with pistil development. Expression of *PySPL14* is associated with the entire floral organ. The expression of *PySPL26* was very low in floral organs at anthesis while it was highly expressed in the middle of flower development, probably because its expression has significantly decreased before the onset of anthesis, so its function on floral organ development needs to be further explored.

## 4. Materials and Methods

### 4.1. Experimental Materials

Samples of flower buds and flower organs were collected in Beijing from January to April 2022 (Figure 7). All samples were frozen immediately in liquid nitrogen and stored at −80 °C for gene expression pattern analysis.

The genome data of *Prunus* × *yedoensis* ‘Somei-yoshino’ was downloaded from GDR (https://www.rosaceae.org/,accessed on 3 March 2022) [47], the genome-wide data of *Arabidopsis thaliana*, and gene sequences of *AtSPL* family of *Arabidopsis thaliana* were downloaded from the *Arabidopsis thaliana* database TAIR (https://www.arabidopsis.org/, accessed on 3 March 2022), the gene sequences and protein sequences of *OsSPLs* family were downloaded from the *Oryza sativa* Genome database RGAP (http://rice.uga.edu/, accessed on 3 March 2022).

### 4.2. Identification and Chromosomal Location of PySPLs

The protein sequences of 16 *AtSPLs* were used as a reference in Genome protein database of *Prunus × yedoensis* ‘Somei-yoshino’ (https://www.rosaceae.org/, accessed on 5 March 2022) for BlastP Searches. Meanwhile, SBP DNA-binding domain (PF031100) from Pfam database (http://pfam.xfam.org/search, accessed on 6 March 2022) was used to identify *PySPLs* by Hmmer Search. Shared candidate *PySPL* sequences of two results were confirmed with the NCBI CDD (https://www.ncbi.nlm.nih.gov/, accessed on 9 March 2022) and SMART program (http://smart.embl-heidelberg.de/, accessed on 9 March 2022) for complete domain analysis [48,49]. Redundant candidates were removed, and the members of the *PySPLs* gene family were finally determined. The protein sequence was submitted to the ExPASy server (https://web.expasy.org/protparam/, accessed on 10 March 2022) to analyze its physicochemical properties and Cell-Ploc2.0 (http://www.csbio.sjtu.edu.cn/bioinf/Cell-PLoc-2/, accessed on 10 March 2022) was used to predict subcellular localization [50]. The location of *PySPLs* on chromosome was analyzed and mapping by TBtools software [51].

### 4.3. Phylogenetic Analysis of PySPLs

The protein sequences of *Prunus × yedoensis* ‘Somei-yoshino’, *Arabidopsis thaliana* and *Oryza sativa* were aligned by the MUSCLE software with the default parameters. The NJ phylogenetic tree was constructed using MEGA11 [52] with 1000 bootstrap replicates 50% partial deletion in Gaps/Missing data treatment. iTOL was used to modify the NJ phylogenetic tree for better visualization [53].

### 4.4. Conserved Domains, Conserved Motifs and Gene Structure Analysis of PySPLs

The MEGA11 software was used to multiple sequence alignment analysis based on the amino acid sequence of *PySPLs* by the neighbor-joining method with 1000 bootstrap replicates and the conserved domain sequences were submitted to the Weblogo3 (http://weblogo.threeplusone.com/, accessed on 13 March 2022) for analysis and visualization [54]. MEME online software (https://meme-suite.org/meme/meme/, accessed on 13 March 2022) was used to confirm conserved motifs in PySPLs protein sequences [55]. Finally, based on the *Prunus × yedoensis* ‘Somei-yoshino’ genome database, the TBtools software was used to analyze the gene structure of all *PySPLs* [51].

### 4.5. Cis-acting Elements Analysis of PySPLs Promoter

Based on the genomic data of *Prunus × yedoensis* ‘Somei-yoshino’, promoter sequences (2000 bp upstream sequences of the start codon) of *PySPLs* were extracted using the TBtools software and submitted to Plantcare (http://bioinformatics.psb.ugent.be/webtools/plantcare/html/, accessed on 16 March 2022) to predict the function and number of *cis*-acting elements. Excel was used to analyze the data and draw the analysis graph [56].

### 4.6. Syntenic Analysis of PySPLs

The duplication pattern of *PySPLs* was analyzed by using One Step MCScanx program of TBtools software and visualized by the Advance Circos package of TBtools [51].

### 4.7. qRT-PCR

To know the expression of *PySPLs* in the flowering induction of *Prunus*
*× yedoensis* ‘Somei-Yoshino’, based on the phylogenetic relationship, conserved motif and gene structure of all *PySPLs*, 12 *PySPLs* were screened out, and primers were designed using Primer premier 5.0 according to CDS sequences (Appendix A), and all primers were checked by primer-Blast of NCBI and primer check of TBtools. The *ACTIN* gene was used as the internal reference. Total RNA was extracted using Rapid Plant RNA Extraction Kit and reverse transcribed into cDNA by Reverse Transcription Kit, all reagents were provided by Aidlab Biotechnologies Co., Ltd. (Beijing, China). qRT-PCR was performed on qTOWER2.2 provided by Analytik Jena AG (Beijing, China) with Takara’s SYBR^®^ Premix Ex TaqTM II provided by Takara (Beijing, China). Each sample was repeated three times. The relative expression of genes was calculated by the 2^−ΔΔCt^ method (1):Relative expression level = 2^−ΔΔCt^
ΔΔCt = ΔCt (test sample) − ΔCt (standard sample)
ΔCt = Ct (Target gene) − Ct (Internal reference gene)(1)
melting curves and standard curves were checked to ensure amplification efficiency. Data processing was performed with Excel, graphs were drawn by GraphPad Prism 9 and significance analysis was performed with ANOVA by SPSS 26, *p* < 0.05 for significance.

## 5. Conclusions

In this study, we identified and analyzed 32 *PySPLs* in the flowering cherry variety *Prunus × yedoensis* ‘Somei-yoshino’ by bioinformatics and conducted a preliminary investigation on its role in the floral transition and flower development of *Prunus × yedoensis* ‘Somei-yoshino’, which provided a genetic resource and a certain basis for further exploring the role of *SPL* transcription factors in the flowering regulation of flowering cherry.

## Figures and Tables

**Figure 1 ijms-23-10052-f001:**
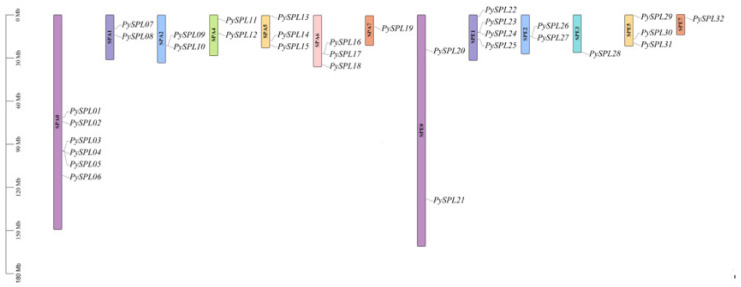
Chromosomal localization of *PySPLs* in *Prun**us ×*
*yedoensis* ‘Somei-yoshino’. The chromosome number is marked in the middle of each chromosome. The scale is in Mega-bases (Mb). Chromosome pairs are marked with the same color.

**Figure 2 ijms-23-10052-f002:**
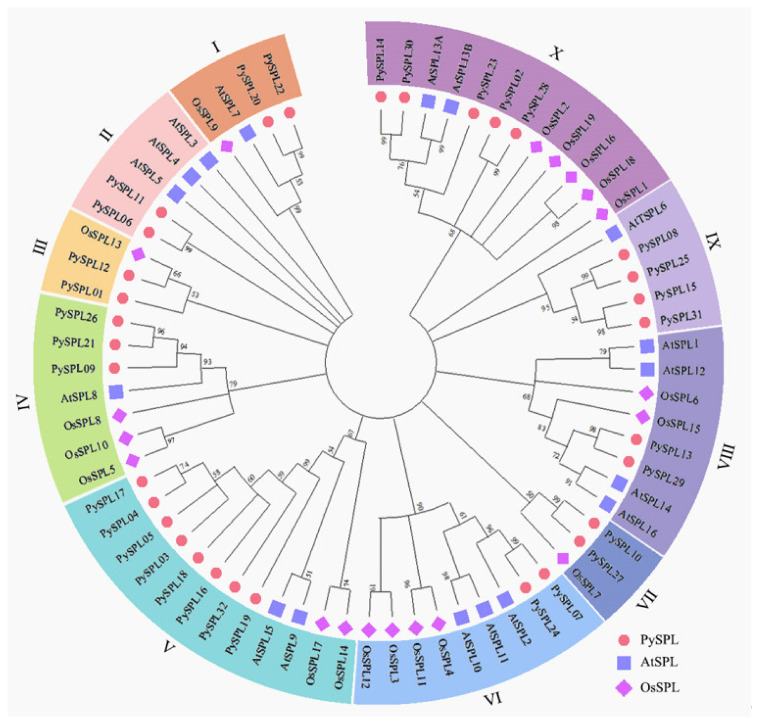
Phylogenetic tree of *SPL* in *Prunus* × *yedoensis* ‘Somei-yoshino’, *Arabidopsis thaliana* and *Oryza sativa.* The full-length amino acid sequences of 32 *PySPLs*, 17 *AtSPLs* and 19 *OsSPLs* was used to construct a phylogenetic tree with 1000 bootstrap replicates by MEGA11. PySPLs were divided into subfamilies according to the branch relationships.

**Figure 3 ijms-23-10052-f003:**
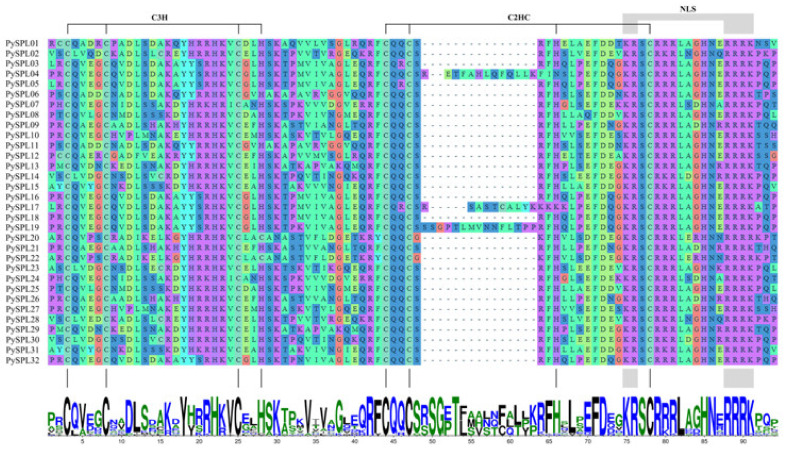
Alignment and conserved domains logo of conserved domains sequence of PySPLs protein.C3H and C2HC were two zinc-finger structures, NLS was bidirectional nuclear localization signal, which were corresponding to the sequence logos of conserved domains.

**Figure 4 ijms-23-10052-f004:**
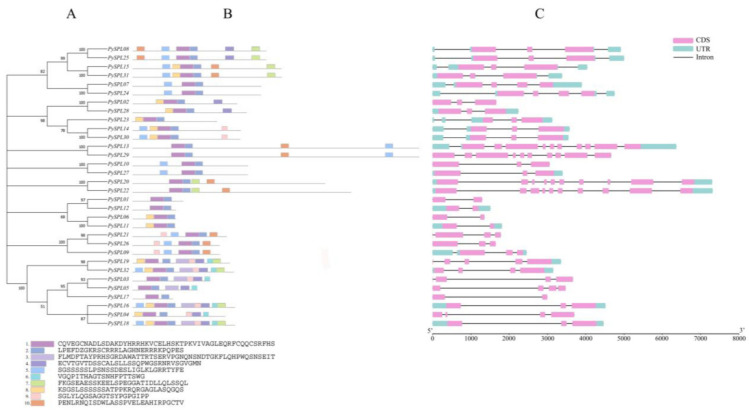
Phylogenetic relationship, conserved protein motif, and gene structure analysis of *PySPLs.* (**A**) A phylogenetic tree harboring amino acid sequence 32 of *PySPLs* by Neighbor-Joining method with 1000 bootstrap replicates. (**B**) Conserved motifs of 32 *PySPLs* arranged according to evolutionary relationships. Motifs of different colors have their corresponding amino acid sequences. (**C**) Gene structure of 32 *PySPLs* arranged according to evolutionary relationships, the exons, introns and UTRs are indicated by different colored boxes and lines.

**Figure 5 ijms-23-10052-f005:**
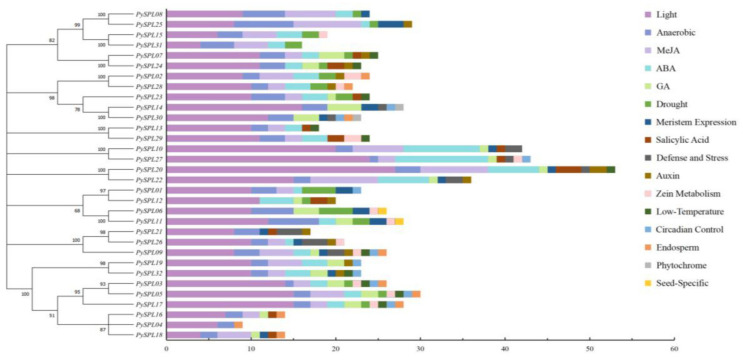
*Cis*-acting elements analysis of *PySPLs* promoters by Plantcare. Different colored boxes represent different *cis*-acting elements. The horizontal coordinate represents the number of *cis*-acting elements.

**Figure 6 ijms-23-10052-f006:**
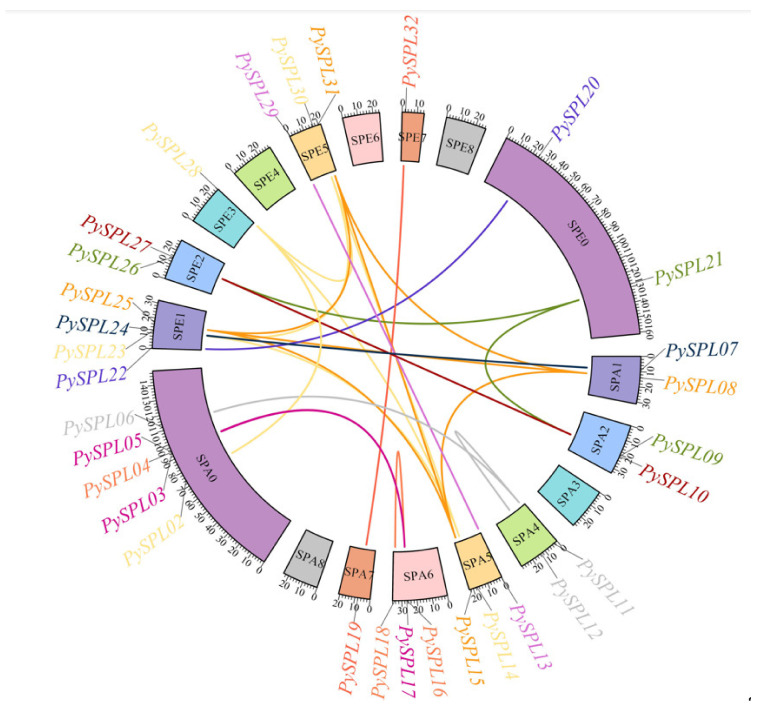
Collinearity analysis of *PySPLs* in *Prunus* × *yedoensis* ’Somei-yoshino’. The chromosome number is marked in the middle of each chromosome. The scale bar of chromosome length is Mb, genes with segmental duplication relationships has the same color markers.

**Figure 7 ijms-23-10052-f007:**
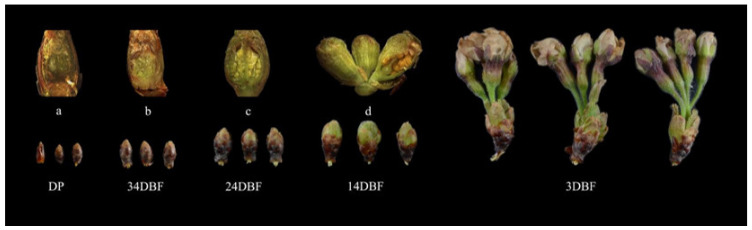
Flower buds of *Prunus × yedoensis* ‘Somei-yoshino’ for qRT-PCR. DP:dormant period, 34DBF:34 day before flowering, 24DBF:24 day before flowering, 14DBF:14 day before flowering,3DBF:3 day before flowering. (**a**–**d**) are the internal morphology of flower bud samples DP, 34DBF, 24DBF, 14DBF, 3DBF respectively.

**Figure 8 ijms-23-10052-f008:**
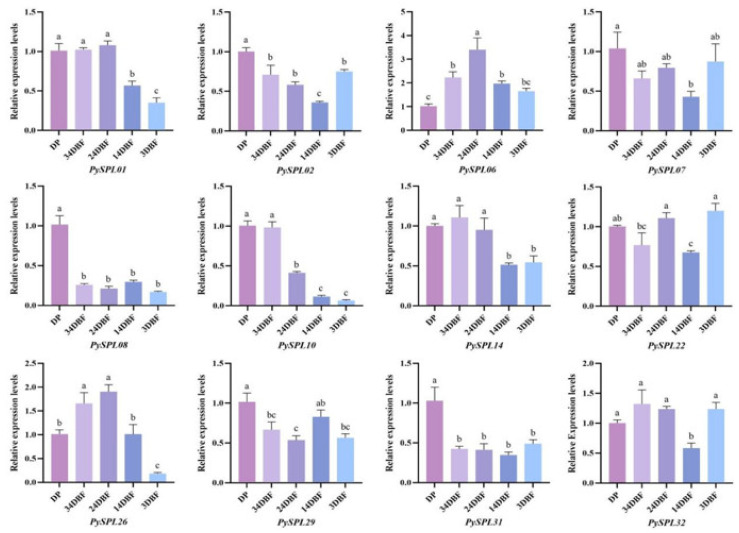
Expression of *PySPLs* in flower buds at different flower developmental stages in *Prunus × yedoensis* ’Somei-yoshino’ by qRT-PCR. DP:dormant period,34DBF:34 day before flowering, 24DBF:24 day before flowering,14DBF:14 day before flowering, 3DBF:3 day before flowering. Each value is shown as average ± standard deviation from three biological replicate sampling. *p* < 0.05 for significance. a, b, c and d represent the level of significant difference between different samples, the difference between samples with the same letter is not significant; the greater the difference of letters, the more significant the difference between samples.

**Figure 9 ijms-23-10052-f009:**
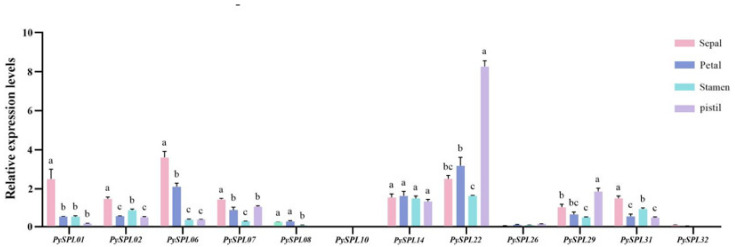
Expression of *PySPLs* in different floral organs in *Prunus* × *yedoensis* ‘Somei-yoshino’. All samples were collected at blooming stage (0DBF). Each value is shown as average ± standard deviation from three biological replicate samplings. *p* < 0.05 for significance. a, b, c and d represent the level of significant difference between different samples, the difference between samples with the same letter is not significant; the greater the difference of letters, the more significant the difference between samples.

**Table 1 ijms-23-10052-t001:** Characteristics properties of *PySPL* in *Prunus×yedoensis* ‘Somei-yoshino’.

Gene	Gene ID	Chromosome	Gene Length (bp)	CDS Length (bp)	Number of Aminocacids	Molecular Wieght (kD)	Theoretical pI	Instability Index	Aliphatic Index	GRAVY	Subcellular Localization
*PySPL01*	CYE_r3.1SPA0_g098740.1	SPA0	1307	570	189	21,562.17	9.26	57.8	53.07	−1.138	Nucleus.
*PySPL02*	CYE_r3.1SPA0_g102580.1	SPA0	1665	1176	391	43,168.56	7.98	59.51	65.14	−0.587	Nucleus.
*PySPL03*	CYE_r3.1SPA0_g124820.1	SPA0	3667	876	291	32,288.48	9.65	48.7	64.60	−0.537	Nucleus.
*PySPL04*	CYE_r3.1SPA0_g125190.1	SPA0	3704	1038	346	37,451.89	9.94	49.04	78.9	−0.256	Cytoplasm. Nucleus.
*PySPL05*	CYE_r3.1SPA0_g125250.1	SPA0	*3475*	732	243	27,329.90	9.64	50.37	63.33	−0.603	Nucleus.
*PySPL06*	CYE_r3.1SPA0_g142750.1	SPA0	1354	489	162	18,589.14	5.75	98.68	36.17	−1.457	Nucleus.
*PySPL07*	CYE_r3.1SPA1_g010660.1	SPA1	3898	1443	480	52,717.28	8.75	42.71	54.48	−0.774	Nucleus.
*PySPL08*	CYE_r3.1SPA1_g017590.1	SPA1	4919	1500	499	55,318.71	7.59	50.89	61.5	−0.654	Nucleus.
*PySPL09*	CYE_r3.1SPA2_g023420.1	SPA2	2457	984	327	36,237.85	8.95	64.88	54.74	−0.788	Nucleus.
*PySPL10*	CYE_r3.1SPA2_g023430.1	SPA2	3059	1296	431	47,188.83	8.07	60.66	56.91	−0.694	Nucleus.
*PySPL11*	CYE_r3.1SPA4_g004730.1	SPA4	1811	486	161	18,419.89	5.92	96.44	36.4	1.456	Nucleus.
*PySPL12*	CYE_r3.1SPA4_g018900.1	SPA4	1510	489	162	18,418.56	9.14	81.42	42.1	−1.156	Nucleus.
*PySPL13*	CYE_r3.1SPA5_g000450.1	SPA5	6368	3213	1070	118,298.50	8.29	55.62	76.48	−0.471	Nucleus.
*PySPL14*	CYE_r3.1SPA5_g020750.1	SPA5	3580	1215	404	44,268.17	8.26	51.4	62.95	−0.687	Nucleus.
*PySPL15*	CYE_r3.1SPA5_g026190.1	SPA5	4040	1671	556	60,834.75	7.05	45.3	67.48	−0.524	Nucleus.
*PySPL16*	CYE_r3.1SPA6_g034400.1	SPA6	4517	1152	383	41,145.63	9.5	55.23	53.73	−0.679	Nucleus.
*PySPL17*	CYE_r3.1SPA6_g034460.1	SPA6	2997	456	152	17,505.46	10.12	60.19	75.72	−0.391	Nucleus.
*PySPL18*	CYE_r3.1SPA6_g047460.1	SPA6	4467	1152	383	41,130.66	9.5	53.79	54.49	−0.662	Nucleus.
*PySPL19*	CYE_r3.1SPA7_g008830.1	SPA7	3356	1098	365	39,954.44	9.68	42.75	50.52	−0.77	Cytoplasm. Nucleus.
*PySPL20*	CYE_r3.1SPE0_g030560.1	SPE0	7304	2160	719	81,272.87	7.2	53.55	78.9	−0.403	Cytoplasm. Nucleus.
*PySPL21*	CYE_r3.1SPE0_g159120.1	SPE0	1785	1056	352	39,548.91	10.07	69	51.16	−0.871	Nucleus.
*PySPL22*	CYE_r3.1SPE1_g001190.1	SPE1	7316	2451	816	91,614.14	6.5	51.94	76.81	−0.42	Cytoplasm. Nucleus.
*PySPL23*	CYE_r3.1SPE1_g014110.1	SPE1	3126	951	316	34,821.23	9.54	60.85	65.19	−0.615	Cytoplasm. Nucleus.
*PySPL24*	CYE_r3.1SPE1_g014420.1	SPE1	4755	1440	479	52,541.15	8.85	45.11	55.41	−0.765	Nucleus.
*PySPL25*	CYE_r3.1SPE1_g021460.1	SPE1	5005	1500	499	55,328.80	7.85	49.39	63.47	−0.621	Nucleus.
*PySPL26*	CYE_r3.1SPE2_g016170.1	SPE2	1650	978	326	36,160.76	8.94	65.78	54.33	−0.776	Nucleus.
*PySPL27*	CYE_r3.1SPE2_g016180.1	SPE2	3397	1296	431	47,228.89	8.07	60.96	57.38	−0.688	Nucleus.
*PySPL28*	CYE_r3.1SPE3_g036000.1	SPE3	2243	1281	426	46,967.87	7.13	60.05	66.41	−0.543	Nucleus.
*PySPL29*	CYE_r3.1SPE5_g000520.1	SPE5	4666	3213	1070	118,346.45	8.21	56.96	75.83	−0.479	Nucleus.
*PySPL30*	CYE_r3.1SPE5_g021220.1	SPE5	3555	1212	403	44,157.07	8.27	51.85	63.35	−0.681	Nucleus.
*PySPL31*	CYE_r3.1SPE5_g026260.1	SPE5	3384	1674	557	61,020.94	6.72	46.49	68.58	−0.511	Cytoplasm. Nucleus.
*PySPL32*	CYE_r3.1SPE7_g002940.1	SPE7	3152	1140	379	41,296.61	9.6	51.09	47.89	−0.811	Nucleus.

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
