# Peer review of "Genome-Wide Analysis of the SPL Gene Family and Expression Analysis during Flowering Induction in Prunus × yedoensis ‘Somei-yoshino’"

_ijms, 2022, doi:10.3390/ijms231710052_

Round 1

Reviewer 1 Report

Manuscript ID: ijms-ijms-1884151

Title:  Genomic Identification of the SPL Gene Family and Expression Analysis during Flowering Induction in Prunus×yedoensis 'Somei-yoshino'

Authors: Gao et al.

General comments: The manuscript deals with the genome wide analysis of SPL gene family in analysis by in silico tools and expression analysis in flower parts and at different development tissues. The manuscript theme is novel however, the content and certain interpretation, analytical details and integration of the present work with previous studies to arrive at some important insights is somewhat lacking and will need more efforts.  In addition, language needs to be improved and gene/protein designations used should be consistent. Expression analysis (a major component of the manuscript) should be appropriately elaborated in the materials and methods (w.r.t. calculations pertaining to the relative expression, primer designing, reverse transcription components, and statistical analysis) and the expression patterns should be carefully analyzed. Introduction and discussion sections need improvement with inclusion of few more appropriate publications relevant to the present manuscript. The common name of the plant should be indicated at the first use in the manuscript. Several concerns are indicated in the PDF file of the manuscript, while the section-specific comments are listed below.

Section specific comments

Title:  Title looks fine but indicates identification and expression analysis. The data however is much more than that and hence change to ‘Genome-wide analysis of the SPL Gene Family and Expression Analysis during Flowering Induction in Prunus × yedoensis 'Somei-yoshino' may be more appropriate.

Abstract: The length of the abstract is fine however the information content and description need to be improved at several places viz. inconsistency in use of gene or protein symbols, more appropriate terms to be used for gene members relationships, expression data interpretation etc. The novelty and overall impact of the work is not getting highlighted.

Keywords: More appropriate keywords should be included

Introduction: The introduction is appropriately w.r.t. length, however background information may be improved, along with description at certain places. Minor changes have been suggested, as indicated in the PDF file, and also listed below:

Lines 33-36:  Statements may be modified to improve the description and content, which may also involve merging the statements to reduce the repetitive content.

Line 37: The gene and protein designations may be re-checked and corrected if needed.

Line 42-43:    Few more roles of SPL and relevant publications may be added.

Lines 44-45:   Some recent literature on SPL genes may be included.

Lines 46-68:   See minor suggestions related to the species name, if used repeatedly.

Lines 69-73:  Indicate the common name along with scientific name when used for the first time. Avoid repetition of same term in a sentence. 

Results:     This section needs to be improved at several places. In general, the results should be written in a concise manner, and data of figure/tables should not be repeated in detail in the manuscript. Suggestions/comments are also highlighted in the PDF file and also listed below.

Line 83:    Section 2.1. heading may need to be changed?

Lines 84-95: Minor changes suggested to improve the content and more clarity of description

Lines 107-112:  The details of phylogenetic analysis may be provided as the protein vary in size, so how an optimal alignment was done for full length proteins should be indicated int he supplementary data. Commonly used terms should be incorporated for the description of clusters/sub-clusters. The extent of divergence may also be indicated and the figure 2 should have bootstrap values as the analysis was carried out but is not evident in the figure. ‘Homology’ in the line 112 may be replaced with’ similarity’

Lines 129-130: Some of the details can be part of the materials and methods.

Line 136-139:  Use correct designations w.r.t. exons or CDS.

Line 151-166:  The description related to the cis-element heterogeneity should be written in a concise manner, and by appropriately minimizing the repetitive information content in the Figure 5 and associated text. Minor corrections are also indicated in the PDF file.

Line 188:    The qRT-PCR expression and statistical significance needs to be carefully interpreted in this section.

Lines 189: Figure 10 should not come before Figure 7 and subsequent figures. If this is important at this place, this may be re-named as Figure 7 and accordingly change names of others.

Line 197:    What was the basis of selection of these 12 genes for qRT-PCR analysis during flower induction is not clear, should be indicated somewhere in the manuscript.

Line 199:  Do the down-regulated genes belong so some specific categories w.r.t. structure or response category.

Line 200:   The PySPL10 gene is included in the previous statement also. See the pattern carefully and rectify the results.

Lines 202-206:  The up- and down-regulation and statistical significance w.r.t. to control should be carefully analysed and described.

Lines 213-214:  All of these listed genes do not show higher levels in most floral organs. The inferences based on the qRT-PCR results should be carefully written.

Lines 218:  Check again the pattern of PsSPL07 and the description? Why some of the very low expressed genes not discussed, as they can be negative regulators.

Discussion:   Discussion section needs to address repetitive content similar to results section and should be more integrated with w.r.t. to previous publications, as well as should focus on the major outcome of the present work.

Lines 229-235:    Some repetition similar to results may be reduced.

Lines 238-239: This statement should be supported with some published references.

Line 241-244:  description similar to results may be minimized.

Lines 254-257:   Content similar to results may be minimized. Add references from previous publications in support.

Lines 271-274:  May not be suitable for discussion looks repetitive as results.

Material and methods: This section is fine, however in certain section sufficient details of methodologies may be provided. In general for most of the analysis (in silico or experiments) sufficient details of key parameters should be provided to enable usage by other labs/researchers.

Section 4.1:    Figure 10 should be carefully placed as indicated in the results section.

Lines 330-331:  The SPL proteins vary in length substantially as per Table 1. How was the alignment done, what parameters were used and whether placement of gaps was appropriate? It will be better to provide details and the alignment file as supplementary data.

Lines 355-356:  More information of primers used for qRT-PCR may be provided. Are primers designed in exon-exon/UTR junctions or not?

Line 360:    How this method was used for estimating the relative expression of genes in Figure 7 is not clear?

Conclusions:      This section is written in very brief manner, and mostly containing repetitive statements. This section might need to be improved after addressing the concerns in the different sections of the manuscript.

Tables: There are few concerns related to the Tables

Table 1 caption may be modified as indicated in the PDF file. Also, Gene designations may be italicized. Information may be updated by including gene length and CDS length before information on encoded proteins

Table 2 can be part of supplementary data

Figures: The concerns related to the Figures are also indicated in the PDF file and some are listed below. Also the figure legends/captions should have been a little bit made more informative w.r.t to key element sin the figures.  

Figure 1: Same colour codes of chromosomes should not be repeated unless there is something to highlight. Minor corrections in Figure caption indicated in the PDF file.

Figure 2: Indicate the bootstrap values and use protein specific designations as italics is generally used to designate gene/transcript.

Figure 3: The Figure caption may be elaborated a little bit to explain key elements of the figure. This logo represents the PySPLs as indicated or only for the domain. use protein specific designations as italics is generally used to designate gene/transcript.

Figure 4: Bootstrap values may be indicated in the NJ tree, and certain sections of the caption may be

Figure 7: Statistical significance of several qRT-PCR data bars and inferences made in terms of up/down-regulation needs to re-checked. On y-axis the title ‘relative expression’ needs to be rectified. Some other suggestions in the caption may also be addressed.

Figure 8: On y-axis the relative expression title needs to be rectified. The 'relative expression' details are not mentioned in the text. Statistical significance of some graphs needs to be re-checked. Why SPL10 transcript is not detected, the gene was expressed in different flowering stages in the previous figure?

Figure 9: The data in the manuscript w.r.t. PySPL06 may not be sufficient to concluded its inferred role. Some more evidences may be needed to support. This figure may be moved to supplementary data.

Figure 10: this figure may be place properly and the time-points for tissue collection/ flower development may be re-designated in terms of number of days from the beginning of the process instead of date of tissue collection.

Reviewer 2 Report

The reviewed manuscript entitled ‘Genomic Identification of the SPL Gene Family and Expression Analysis during Flowering Induction in Prunus×yedoensis 'Somei-yoshino'’ written by Lan Gao et al. presents interesting results concerning the genetic aspects of Prunus×yedoensis flowering. The authors analyzed genes of the PySPL family and evaluated their association with flower development using bioinformatics and gene expression experiments. The change in the expression of PySPL06 and other genes was related to the development of different parts of Prunus×yedoensis flowers, providing new information about the genetic regulation of flower development in this plant. The article is well structured and scientifically sound. I have only minor comments on this manuscript.

1. The authors provided knowledge about regulation of flower development in Prunus×yedoensis, but it is not clear whether obtained results have a practical value and could be used, for example, to modulate the flower development in the studied plant. If yes, please address this aspect in the manuscript.

2.   In the text there are editorial errors, including additional spaces (like in line 111), lack of spaces between words (like in line 119), and non-consequent italizing (like in line 25). Furthermore, check the formatting of the references list to fit it to the journal requirements.

3. In the Abstract section, the abbreviations PySPL and SBP should be explained to make the text understandable for people who are not specialists in the topic.

4.    Line 83 – title of 2.1 paragraph has a gap, please check it.

5. For the readers’ convenience, figure 10 should be placed in the paragraph 2.8, where is referred for the first time. Furthermore, all time points presented on this figure should be explained in the figure legend similarly as in the text of paragraph 2.8.

6. It is unclear what criteria were used to select 12 PySPL genes for expression experiments, out of the whole group of 32 genes.

7. In the legend of figure 8, add the information in which time point the flower parts were collected.

8. In the 4.7 paragraph, add information about manufacturers of used reagents and equipment, used statistical tests for calculation of statistical significance and how high amplification efficiency was ensured as required for 2-ddCt method.

Round 2

Reviewer 1 Report

Manuscript ID: ijms-1884151 (revised version)

The revised version of the manuscript has addressed all the concerns, suggestions and comments of the Reviewer's, appropriately. The updated version has been considerably improved in terms of information content, technical details and presentation.